# A Systematic Review and Meta-Analysis of the Clinical Use of Megestrol Acetate for Cancer-Related Anorexia/Cachexia

**DOI:** 10.3390/jcm11133756

**Published:** 2022-06-28

**Authors:** Yu Liang Lim, Seth En Teoh, Clyve Yu Leon Yaow, Daryl Jimian Lin, Yoshio Masuda, Ming Xuan Han, Wee Song Yeo, Qin Xiang Ng

**Affiliations:** 1MOH Holdings Pte Ltd., 1 Maritime Square, Singapore 099253, Singapore; yulianglim95@gmail.com; 2Yong Loo Lin School of Medicine, National University of Singapore, 10 Medical Dr, Singapore 117597, Singapore; e0659260@u.nus.edu (S.E.T.); e0268630@u.nus.edu (C.Y.L.Y.); daryllinjimian@gmail.com (D.J.L.); y.masuda@u.nus.edu (Y.M.); 3Department of Paramedicine, Monash University Peninsula Campus, Frankston, VIC 3199, Australia; mxhan9598@yahoo.com; 4Mount Elizabeth Hospital, 3 Mount Elizabeth, Singapore 228510, Singapore; corneliuslionel@gmail.com

**Keywords:** megestrol acetate, megace, anorexia, cancer, palliative, quality of life

## Abstract

Cancer-related anorexia/cachexia is known to be associated with worsened quality of life and survival; however, limited treatment options exist. Although megestrol acetate (MA) is often used off-label to stimulate appetite and improve anorexia/cachexia in patients with advanced cancers, the benefits are controversial. The present meta-analysis aimed to better elucidate the clinical benefits of MA in patients with cancer-related anorexia/cachexia. A systematic search of PubMed, EMBASE, OVID Medline, Clinicaltrials.gov, and Google Scholar databases found 23 clinical trials examining the use of MA in cancer-related anorexia. The available randomized, controlled trials were appraised using Version 2 of the Cochrane risk-of-bias tool (RoB 2) and they had moderate-to-high risk of bias. A total of eight studies provided sufficient data on weight change for meta-analysis. The studies were divided into high-dose treatment (>320 mg/day) and low-dose treatment (≤320 mg/day). The overall pooled mean change in weight among cancer patients treated with MA, regardless of dosage was 0.75 kg (95% CI = −1.64 to 3.15, τ^2^ = 9.35, I^2^ = 96%). Patients who received high-dose MA tended to have weight loss rather than weight gain. There were insufficient studies to perform a meta-analysis for the change in tricep skinfold, midarm circumference, or quality of life measures. MA was generally well-tolerated, except for a clear thromboembolic risk, especially with higher doses. On balance, MA did not appear to be effective in providing the symptomatic improvement of anorexia/cachexia in patients with advanced cancer.

## 1. Introduction

As a result of various central and peripheral causes including a greater inflammatory response, many patients with advanced cancers experience a marked loss of appetite, loss of weight, asthenia, and a poor prognosis [1,2,3]. This is collectively referred to as the cancer anorexia/cachexia syndrome, and it happens in more than half of all cancer patients [3]. Sustained loss of appetite and/or an aversion to food often compounds emotional distress in both patients and their caregivers [4] and are admittedly difficult aspects of cancer for patients’ loved ones to comprehend [4,5].

Cancer-related anorexia/cachexia is also clinically significant as a patient’s nutritional status affects their quality of life [6] and overall prognosis [7]. The weight loss of more than 5 percent of premorbid weight prior to the initiation of chemotherapy is associated with increased morbidity and early mortality [7].

Providing dietary counseling, nutritional support and nutritional therapies are therefore important and endorsed by major clinical practice guidelines [8]. However, options may be limited as cancer-related cachexia is also often refractory to conventional nutritional support [9]. The management of cancer-related anorexia remains a substantial clinical challenge and numerous off-label, pharmacologic therapies have been tried, with variable tolerability and dissimilar efficacy on clinical outcomes and the quality of life measures [10,11,12]. One such example is megestrol acetate (MA), a synthetic progestin, which is often used to boost appetite and body weight in patients with cancer cachexia [12,13]. In clinical studies, MA has been found to decrease circulating inflammatory cytokines [13] and stimulate increases in body mass [14].

However, a 2013 Cochrane review [12] and 2018 systematic review [15] yielded inconclusive findings regarding the efficacy of MA for the treatment of anorexia/cachexia syndrome. Furthermore, the 2018 systematic review had marked heterogeneity and included patients with anorexia/cachexia related to any pathology (e.g., cancer, acquired immunodeficiency syndrome (AIDS), etc.). The optimal dosing strategy for MA also remains unknown. Given that newer randomized, controlled trials [16,17] have been published since, this updated systematic review and meta-analysis is thus timely and necessary. 

## 2. Methods

This systematic review and meta-analysis adhered to the Preferred Reporting Items for Systematic Reviews and Meta-Analyses (PRISMA) guidelines [18]. The study protocol was registered in the International Prospective Register of Systematic Reviews (PROSPERO), registration number CRD42022320128.

### 2.1. Search Strategy

A systematic literature search was performed in accordance with the latest Preferred Reporting Items for Systematic Reviews and Meta-Analyses (PRISMA) guidelines [18]. By using the following combinations of broad Major Exploded Subject Headings (MesH) terms or text words [megestrol] AND [anorexi* OR cachex* OR cachectic OR weight OR appetite], a comprehensive search of PubMed, EMBASE, OVID Medline, Clinicaltrials.gov, and Google Scholar databases yielded 2942 papers published in English between 1 January 1988 and 1 May 2021. Attempts were made to search the grey literature using the Google search engine. The titles and abstracts of records were downloaded and imported into EndNote bibliographic software and from there to the Covidence online tool (Vertitas Health Innovation Ltd, Melbourne, Australia. Available at www.covidence.org) to streamline our systematic review process. All duplicates were automatically removed once uploaded to Covidence. Titles and abstracts from the preliminary search were retrieved and reviewed for relevance independently by two study investigators (Q.X.N. and Y.L.L.). Full articles of relevant studies were then retrieved for further review and assessed by three study investigators (Q.X.N., Y.L.L., and M.X.H.) for inclusion based on the pre-defined criteria. All retrieved publications were manually reviewed and also checked for references of interest. Discrepancies were resolved by consensus amongst the three study investigators (Q.X.N., Y.L.L., and M.X.H.).

### 2.2. Inclusion and Exclusion Criteria

The inclusion criteria for this review were: (1) randomized, controlled trial (RCT); (2) study population involving oncological patients; (3) had cancer-related anorexia or cachexia as a primary endpoint; and (4) reported outcome measures on weight and/or quality of life. Any disagreement on inclusion was resolved by consensus. Exclusion criteria included cohort studies, single case reports or case series, conference abstracts, and proceedings, which were not accepted for this review.

### 2.3. Data Abstraction

Data were extracted using a standardized electronic form. Each article was double-coded by either pair of researchers (C.Y.L.Y./S.E.T. or Y.M./D.J.L.), blinded within pairs. Disputes were resolved through consensus from the senior author (Q.X.N.). Data abstracted included the study characteristics (e.g., author name, year of publication, and country) and study population characteristics (e.g., sample size, country, study population, dosage of MA). The dosages of MA treatment were dichotomized into high dosage (>320 mg/day) and low dosage (≤320 mg/day). The primary outcomes collected were the change in weight (in kg), quality of life improvement, and side effects experienced for the duration that patients were treated with MA.

For continuous variables, the mean and standard deviation (SD) were abstracted. Where these data were unavailable, appropriate formulae were applied to transform the data from the median and range or interquartile range to the mean and SD. In the event where SD was unable to be derived from the aforementioned formulae, another formulae was used to derive the SD from other included studies.

### 2.4. Statistical Analysis 

Data analyses were performed using R 4.0.3 (R Foundation for Statistical Computing, Vienna, Austria). A single-arm meta-analysis of means was conducted to pool the mean change in weight, tricep skinfold, and midarm circumference of patients who received megestrol acetate treatment. Individual studies were weighted by the inverse variance method. Heterogeneity was quantified using the τ^2^ and I^2^ statistics. I^2^ value thresholds of 25%, 50%, and 75% signified low, moderate, and high heterogeneity, respectively. All models were random effects, regardless of the statistical heterogeneity. This was conducted as we expected clinical heterogeneity arising from different populations and time points. Two-tailed statistical significance was set at a *p*-value ≤ 0.05. Funnel plots, Egger regression test, and the Begg and Mazumdar rank correlation test were performed to evaluate the publication bias only when there were at least 10 data points.

For data that had fewer than three data points, meta-analysis was considered to be inappropriate and they were instead systematically reported. Quality of life improvement and the side effects experienced due to treatment with megestrol acetate treatment were also systematically reported.

### 2.5. Risk of Bias Assessment

The risk of bias assessment was conducted using Version 2 of the Cochrane risk-of-bias tool for randomized trials (RoB 2) [19]. The RoB2 tool assesses the quality on five domains: the randomization process, deviations from intended interventions, missing outcome data, outcome measurements, and reporting, graded based on the consensus of three study investigators (Q.X.N., Y.L.L., and M.X.H.). 

## 3. Results

### 3.1. Retrieval of Studies

Figure 1 detailed the study selection and identification process. A total of 2942 records were found from the database search with 1842 records marked ineligible by automated filters and 368 records removed as duplicates. A total of 675 articles were further excluded after title and abstract screening, and subsequently, 34 articles were excluded after the full text review. Finally, a total of 23 studies were systematically reviewed, albeit only eight contained sufficient anthropometric data to perform a meta-analysis.

The 23 included studies represented a total of 3790 cancer patients treated with MA, and originated from seven countries, namely Australia, Canada, China, Italy, Taiwan, United Kingdom, and the United States of America. The study sample sizes ranged from six to 475 and the study duration was eight months maximum. The characteristics of the included studies are further described in Table 1.

### 3.2. Meta-Analysis of Pooled Mean Change in Weight 

A total of eight studies provided sufficient data on the weight change. The overall pooled mean change of weight among cancer patients treated with megestrol acetate, regardless of dosage was 0.75 kg (95% CI = −1.64 to 3.15, τ^2^ = 9.35, I^2^ = 96%) (Figure 2). For the purposes of the meta-analysis, the dosages of the MA treatment were dichotomized into high dosage (>320 mg/day) and low dosage (≤320 mg/day).

The pooled mean change of weight among the cancer patients treated with high-dose megestrol acetate was −0.05 kg (95% CI = −2.71 to 2.60, τ^2^ = 5.26, I^2^ = 94%) (Figure 2). The pooled mean change of weight among cancer patients treated with low-dose megestrol acetate was 2.24 kg (95% CI = −7.19 to 11.67, τ^2^ = 9.35, I^2^ = 96%) (Figure 2). In all instances, the SMD did not achieve statistical significance. 

There were insufficient studies (<3) to perform a meta-analysis for change in tricep skinfold and midarm circumference with megestrol treatment.

### 3.3. Risk of Bias Assessment

The included RCTs were appraised using the RoB2 and were classified to be of moderate-to-high risk of bias. The detailed risk of bias assessment results are available in Appendix A.

### 3.4. Publication Bias Assessment

There was no evidence of publication bias, based on a non-significant Egger regression test (*p* = 0.858) and Begg and Mazumdar rank correlation test (*p* = 0.621) and a visually symmetrical funnel plot (Figure 3).

## 4. Discussion

Despite the prevalence, the etiology of cancer-related anorexia/cachexia is incompletely understood but probably multifactorial in nature. Overall, MA did not appear to improve the weight gain amongst patients with cancer-related anorexia/cachexia. Notably, the high-dose MA also seemed to produce weight loss rather than weight gain when compared with the low-dose MA. However, this could be due to the fact that patients who received higher doses of MA may have had more refractory cachexia. In the study by [36], forty-six (63%) of the patients with advanced gastrointestinal cancer did not complete the trial as they had worsened disease, requiring further supportive care or pain control.

Based on a systematic review of available evidence, MA also did not appear to improve quality of life although limited studies examined this. In terms of the potential adverse events associated with its use, MA was generally well-tolerated, except for a clear thromboembolic risk, especially with higher doses. 

In terms of the biological mechanisms of MA, it is a synthetic progesterone and may act to stimulate appetite and increase the body fat stores, but not lean body mass [42]. The metabolic effects are likely mediated via its anti-inflammatory actions. Studies have noted that after MA was discontinued, the effects were not sustained and weight loss reverted [43]. As with other progestins, common side effects would include headaches and nausea, and high doses sometimes cause thrombosis. 

The findings of the present meta-analysis significantly extend those of earlier meta-analyses [12,15]. Compared to an earlier meta-analysis by Ruiz-Garcia et al. [15], which included patients with AIDS, anorexia nervosa, degenerative diseases, and other terminal illnesses, we focused specifically on patients with cancer-related anorexia/cachexia. We also included several studies [20,22,23,24,25,27,29,31,36,37,38,39,40,41] that were missed in the earlier 2018 review, and incorporated the findings of a recent randomized, double-blind, placebo-controlled RCT [17]. The 2018 review also did not provide any relevant changes in the MA effectiveness compared to the 2013 Cochrane review [12]. The present meta-analysis provides us with greater surety in recommending against the use of megestrol acetate for the symptomatic improvement of anorexia/cachexia in oncological patients with advanced cancer. The benefits of MA use were based on only low-quality evidence and MA did not produce a significant weight gain or notable improvements in the quality of life measures.

### Limitations

Limitations of the present meta-analysis include that the literature in this field was generally dated, with the majority of the literature (13 of 23 included studies) on MA use in cancer-related anorexia/cachexia published more than 15 years ago. Moreover, there was considerable heterogeneity amongst the included studies, with patients with different malignancies and at different stages of the disease including those who were actively dying (i.e., refractory cachexia). Gastrointestinal cancers and metastases may also produce more profound anorexia/cachexia than those elsewhere because of the obstruction of the digestive tract. Second, the available trials were not designed with sufficient power to detect clinically meaningful differences in adverse events or survival. Third, there was also no information regarding the potential long-term benefits and harms associated with MA use given the limited study duration (up to 8 months).

## 5. Conclusions

MA did not produce significant weight gain in patients with advanced cancers. There was also no difference between patients who received high-dose (>320 mg/d) and low-dose (≤320 mg/d) MA. MA also did not appear to be associated with improvements in quality of life measures although limited studies were available for meta-analysis. On balance, the routine use of MA for cancer-related anorexia/cachexia should not be recommended, although there may be benefits in specific patient subpopulations, and this should be the focus of future research.

## Figures and Tables

**Figure 1 jcm-11-03756-f001:**
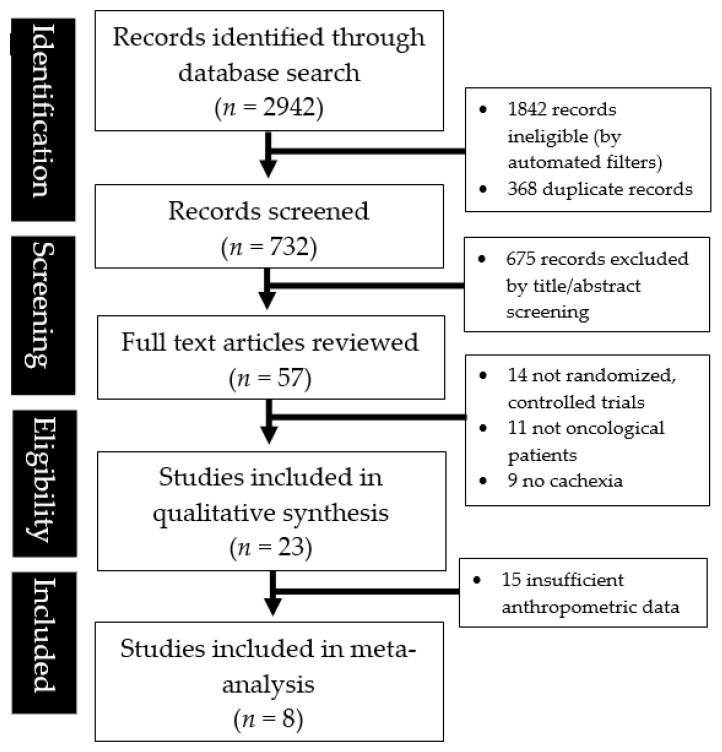
The PRISMA diagram illustrating the literature search and abstraction process.

**Figure 2 jcm-11-03756-f002:**
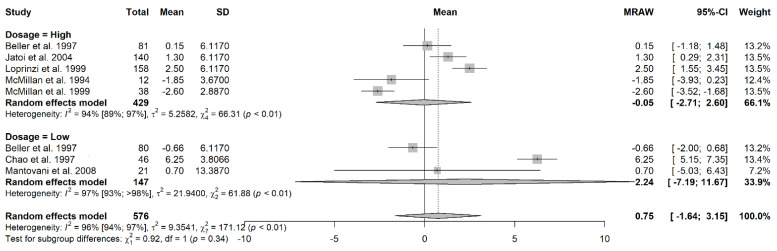
The forest plot showing the pooled mean change in weight for patients who received megestrol acetate treatment [21,22,30,32,34,35,36].

**Figure 3 jcm-11-03756-f003:**
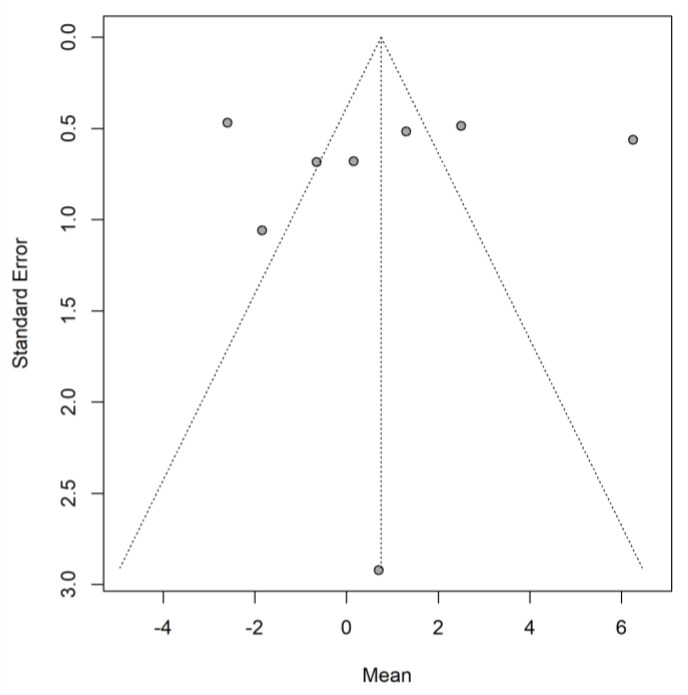
The funnel plot of studies that provided sufficient data on weight change.

**Table 1 jcm-11-03756-t001:** The characteristics and findings of the studies reviewed (arranged in alphabetical order according to the first author’s last name).

Author/Year	Country	Study Design (N)	Study Population	Intervention(s)	Primary and Secondary Endpoints	Key Findings
Abrams et al., 1999 [20]	United States	Randomized, controlled trial (*n*= 368)	Females only; age ≥18 years; histologically documented breast cancer and progressive metastatic; prior usage of progestins not allowed; chemotherapy for metastatic disease not allowed.	MA 160 mg/day, or 800 mg/day or 1600 mg/day	Response rate; response duration; time to disease progression; weight gain; overall survival	**Weight gain:** MA 160 mg/day = 37% of patients reported a 5% weight gain and 2% of patients reported a 20% weight gain; MA 800 mg/day = 70% of patients reported a 5% weight gain and 23% of patients reported a 20% weight gain; MA 1600 mg/day = 66% of patients reported a 5% weight gain and 20% of patients reported a 20% weight gain. **Overall survival:** Increased MA dose did not affect survival duration. **Side effects:** Serious (grades 4 and 5) thrombotic events increased with MA dose: MA 160 mg/day = 4; MA 800 mg/day = 3; MA 1600 mg/day = 6.
Beller et al., 1997 [21]	Australia	Randomized, double-blind, placebo-controlled trial (*n* = 240)	Endocrine-insensitive cancer; body weight at least 5% below ideal or unintentional loss of at least 5% usual body weight.	MA 160 mg/day or 480 mg/day, or placebo	Quality of life measures; nutritional status (weight, mid-arm circumference, triceps skinfold thickness, serum albumin); survival time	**Weight gain:** No significant difference in weight change between treatment groups (*p* = 0.29); placebo −0.15 kg; MA 160 mg/day −0.66 kg; MA 480 mg/day +0.15 kg. **Quality of life:** Combined quality of life measures showed a significant improvement in the MA 480 mg/day group (*p* < 0.001). No evidence of a time effect. Patients who received MA reported substantially better appetite (*p* = 0.001), mood (*p* = 0.001), and overall quality of life (*p* < 0.001), and possibly less nausea and vomiting (*p* = 0.08) compared to the patients receiving the placebo. **Side effects:** 2 (2.5%) in the 160 mg/day group had pulmonary emboli; 4 (5%) in the 160 mg/day group had severe edema.
Chao et al., 1997 [22]	Taiwan	Open phase II trial (*n* = 46)	Pathologically confirmed inoperable or metastatic HCC.	MA 160 mg/day	Clinical benefit; survival; weight gain; quality of life	**Weight gain:** 14 of 22 (63.6%) patients had an increase in lead body weight; median increase of 5 kg, range 1 to 14 kg (mean 6.25, SD 3.8066). **Clinical benefit:** 13 patients had an increase in serum albumin (median 0.3 g/dL, range 0.1 to 1 g/dL). **Quality of life:** 20 of 32 (62.5%) patients reported improvements in appetite and feeling of well-being. **Side effects:** 1 patient had mild congestive cardiac failure (3.13%), 1 hyperglycemia (3.13%) and 9 had mild edema (28.1%).
Chow et al., 2011 [23]	Myanmar, New Zealand, Philippines, Singapore, South Korea, Vietnam	Randomized, double-blind, placebo-controlled trial (*n* = 204)	Patients with advanced HCC.	Placebo or MA 320 mg/day	Survival; quality of life; side effects	**Quality of life:** Placebo had more favorable quality of life although MA had favorable improvements in levels of appetite loss and nausea/vomiting episodes. **Side effects:** Similar between groups.
Collichio et al., 1998 [24]	United States	Open phase II trial (*n*= 15)	Patients with renal cell carcinoma; Eastern cooperative oncology group performance status of ≤2.	Interferon alpha-2b, 10 million IU/m^2^ and MA 160 mg/day	Clinical benefit; weight gain; side effects	**Weight gain:** 7 of 11 patients lost weight. **Side effects:** 12 patients (80%) had fatigue, 9 (60%) had flu-like symptoms and 3 (20%) had nausea/vomiting.
Couluris et al., 2008 [25]	United States	Open clinical trial (*n* = 6)	Pediatric patients (between ages 2 and 20 years), with diagnosis of cachexia secondary to cancer or cancer treatment; exhibited no response to Cyproheptadine hydrochloride (CH).	CH at 0.25 mg/kg/d orally in 2 divided doses, and MA 10 mg/kg/d in a single daily oral dose	Clinical benefit; weight gain; side effects	**Weight gain:** The average weight gain was 2.5 kg (range: 0.6 to 5.9 kg). **Side effects:** 1 patient (16.7%) had asymptomatic hypocortisolemia and hyperlipidemia.
Currow et al., 2021 [17]	Australia	Randomized, double blind, placebo-controlled trial (*n* = 190)	Patients with advanced cancer and known to a palliative care team; were mentally competent; able to take oral medications; had a baseline appetite score of ≤4 on a 0–10 numeric rating scale (where 0 is no appetite and 10 is best possible appetite); and an Eastern Cooperative Oncology Group (ECOG) score of 0–3; or Australia-modified Karnofsky performance status (AKPS) score of 30–100.	MA 480 mg/day, or dexamethasone 4 mg/day, or Placebo	Quality of life scores; weight, appetite scores, AKPS; side effects	**Weight:** There were no differences in weight stability between groups (*p* = 0.2417). MA = 87% responded; Dex = 74% responded; Placebo = 85% responded. **Quality of life:** 79.3% (*n* = 48) of participants in the megestrol group, 65.5% (*n* = 44) in the dexamethasone group and 58.5% (*n* = 36) in the placebo group had at least 25% improvement of appetite score. No significant difference between treatment arms including quality of life. **Side effects:** Generally well-tolerated.
Cuvelier et al., 2014 [26]	Canada	Randomized, controlled trial (*n* = 26)	Patients <18 years; cancer diagnosis; weight loss secondary to cancer or cancer treatment (must have lost ≥5% body weight from previous recorded weight or have history of anorexia); able to tolerate orally and have life expectancy of at least 3 months.	MA suspension (7.5 mg/kg/day, maximum 800 mg/day), or placebo	Nutritional status (weight, height, mid upper arm circumference, triceps skin fold thickness, body composition analysis); blood glucose levels; 8am cortisol levels	**Weight gain:** MA group experienced 19.7% ± 15.3% weight gain, while placebo group had −1.2% ± 4.9% weight change. Statistically significant difference in mean percent weight change of +20.9% in favor of MA over placebo was observed (*p* = 0.003). **Nutritional values:** Patients in the MA group experienced an increase in mean weight-for-age z-score (WAZ) of +1.00% (±0.79%). Patients in the placebo group experienced a decrease in mean WAZ of −0.18% (±0.34%). Statistically significant difference in mean WAZ of +1.18% in favor of MA over placebo was observed (*p* = 0.002). BMI-for-age z-scores were higher in the MA group. Patients in the MA group experienced an increase in mean BMI-Z of +1.58% (±1.37%), while patients in the placebo group experienced a decrease in mean BMI-Z of −0.29% (±0.50%). **Side effects:** severe (15%, *n* = 2) and mild (15.4%, *n* = 2) adrenal suppression in the MA group.
Greig et al., 2014 [27]	United Kingdom	Open phase I/II trial (*n* = 13)	Adult patients with advanced cancer and cachexia	Formoterol 80 μg/day and MA 480 mg/day	Muscle response; body weight; physical activity; quality of life measures; side effects	**Weight gain:** 5 out of 7 patients increased their total body weight, with study’s overall mean total body weight increased by 2.6% from 58.7 kg at baseline to 60.2 kg at 8 weeks (*p* = 0.379). **Clinical benefit:** 6 had major muscle response (magnitude of strength or muscle size improvement). **Quality of Life:** QLQ-C30 questionnaire was unchanged, functional domains all showed a small, non-significant increase. For anorexia, the improvement in symptoms was marked (*p* = 0.005). **Side Effects:** Most common being tremor (8 reports in 7 patients), tachycardia. 4 discontinued, possibly related to side effect.
Guo et al., 2002 [28]	China	Randomized control trial (n = 92)	Patients with stage IIIb–IV non-small cell lung cancer, judged to be endurable ≥3 cycles of chemotherapy, no other significant comorbidities, Karnosfsky ≥50, with life expectancy ≥3 months.	IV Arsenous acid and IV Tα-1 thymus peptide and MA 160 mg/day, or chemotherapy in NP protocol (Navelbine and Cisplatin)	Tumor response; Karnofsky score; body weight; T cell subset and NK cell activity; side effects	**Weight gain:** 33% of treatment group gained >7% body weight, as compared to 12.5% of control group (*p* <0.05). **Clinical benefit:** 44% of treatment group had Karnofsky score increased by ≥20 as compared to 20% of control group (*p* < 0.05). Showed no significant difference in tumor therapeutic response or survival rate. **Side Effects:** Treatment group had higher reported rates of aversion to cold, fever, pantalgia, and stuffy nose compared to control group.
Jatoi et al., 2002 [29]	USA	Randomized, double blind, controlled trial (*n* = 469)	Adult patients with incurable malignancy (excluding brain, breast, ovarian, endometrial) with estimated life expectancy ≥3 months, ECOG score 0–2, and prior weight loss of at least 2.3 kg or intake <20 calories/kg/day, and a belief that they have anorexia or weight loss.	MA 800 mg/day and placebo, or Dronabinol 5 mg/day and placebo, or combination of MA and Dronabinol	Improvement of appetite; weight gain; quality of life	**Weight gain:** MA group have showed that there’s significant difference in weight gain (23% have 1–4% weight gain, 10% have 5–9% gain, 10% have ≥10% gain) as compared to the Dronabinol group (23% have 1–4% gain, 8% have 5–9% gain, 3% have ≥10% gain) (*p* = 0.041). **Quality of Life:** 75% of the MA group, as compared to 49% of the Dronabinol group (*p* = 0.0001) reported improvement in appetite. Among all 3 study arms, there are no significant differences between the maximal score improvement of QoL Uniscale assessment (MA: 15 ± 19, Dronabinol: 12 ± 8 [*p* = 0.19], Combination 14 ± 19 [*p* = 0.72]). For FAACT scores, the MA group (10.3 ± 11) showed that there was significant difference between baseline and maximum score compared to Dronabinol group (7.2 ± 10, *p* = 0.002), but no significant difference compared to combination group (9 ± 10, *p* = 0.30). **Side Effects:** 18% of the male participants in MA group reported significant presence of impotence as compared to 4% with Dronabinol (*p* = 0.002). Other noted side effects that were not significant compared to other treatment arms included vomiting, fluid retention, muddled thinking, drowsiness, loss of coordination, and inappropriate behavior.
Jatoi et al., 2004 [30]	USA + Canada	Randomized, double blind, controlled trial (*n* = 421)	Adult patients with incurable malignancy (excluding brain, breast, ovarian, endometrial) with estimated life expectancy ≥3 months, ECOG score 0–2, and prior weight loss of at least 2.3 kg or intake <20 calories/kg/day, and a belief that they have anorexia or weight loss.	MA 600 mg/day and placebo, or eicosapentaenoic acid (EPA) supplement 2 cans/day and placebo, or combination of both MA and EPA	10% weight gain above baseline, improvement in appetite, survival, quality of life, side effects	**Weight gain:** 18% of the MA group showed >10% gain as compared to 6% of EPA group and 11% of combination group (*p* = 0.01). Separate analysis in mean weight changes showed significant changes across all arms (*p* = 0.03), where mean weight gain of MA is 1.3 kg, EPA is 1.0 kg, and combination is 0.1 kg. **Clinical benefit:** No statistically significant differences in median survival (*p* = 0.82) across all 3 groups. **Quality of life:** Appetite measured using NCCTG was comparable across all treatment arms, all with favorable effects–EPA: 64%, MA: 68%, Combination: 66% (*p* = 0.69). In terms of quality of life assessment based on difference of maximal and baseline Uniscale score, it showed no significant differences across all groups–median changes 0 in EPA, 0 in MA, 1 in combination (*p* = 0.93). **Side effects:** Apart from impotence in 9% of MA group, 3% of EPA group, 19% in combination group (*p* = 0.006), other side effects including nausea, vomiting, confusion, swelling of legs, and thromboembolism were not significantly different.
Levitan et al., 1998 [31]	USA	Phase II trial (*n* = 30)	Adult patients with stage IIIB or IV NSCLC, deemed inoperable, with no prior cytotoxic chemotherapy, ECOG status of 0–1, and life expectancy of ≥12 weeks, no serious comorbidities that would interfere chemotherapy.	Cisplatin 50 mg/m^2^, Ifosfamide 2 g/m^2^, Mesna, 7-day course oral etoposide on days 1, 15, 29, 43, and 57, followed by 7-day filgrastim after each course of etoposide, along with MA 250 mg/daily throughout duration of 10 week therapy	Weight change, clinical response, survival, toxicity	**Weight change**: Paired comparisons of pre- and post-treatment weights show no difference.
Loprinzi et al., 1999 [32]	USA	Randomized, controlled trial (*n* = 475)	Patients with incurable cancer (other than breast, prostate, ovarian, endometrial), with a history of losing at least 5 pounds within the previous 2 months or have an estimated daily caloric intake of <20 cal/kg, expected life expectancy of at least 3 months and ECOG performance of at least 2, no evidence of ascites, obstructive or functional alimentary tract issues, not receiving supplementary feeds.	MA 800 mg/day, or Fluoxymesterone 20 mg/day, or Dexamethasone 3 mg/day	Weight gain, appetite improvement, side effects	**Weight gain:** Weight gain of >10% from baseline was greater for MA compared to fluoxymesterone. (*p* = 0.08) No difference in MA vs. dexamethasone (*p* = 0.42). No difference in all 3 arms when comparing median and mean maximal weight gain from baseline. **Quality of life:** MA is found to have a significant increase in appetite when compared to fluoxymesterone, and similar efficacy when compared to dexamethasone. Average maximum, quality of life values per patient were 67, 71, 69 for MA, dexamethasone, and fluoxymesterone, with no statistical significance. **Side Effects:** Commonly includes myopathy, infection, and thromboembolic disease. Thromboembolic phenomenon was present in 5%, 2%, and 1% of MA, fluoxymesterone, dexamethasone treatment arms, respectively.
Maddedu et al., 2012 [33]	Italy	Phase III, randomized controlled trial (*n* = 60)	Patients (aged 18–85 years) with advanced stage tumor at any site, loss of ≥5% of ideal or pre-illness body weight in the last 6 months and a life expectancy of ≥4 months.	L-carnitine 4 g/day and Celecoxib 300 mg/day, or L-carnitine 4 g/day and celecoxib 300 mg/day and megestrol acetate 320 mg/day	Increase in Lean Body Mass (LBM), improvement of total daily physical activity, physical performance, fatigue, resting energy expenditure, ECOG status, Glasgow prognostic score, proinflammatory cytokines, appetite, quality of life	**Weight gain:** Lean body mass has no significant differences between the two arms. MA arm showed a trend towards increase in body weight (*p* = 0.052) **Clinical benefit:** Changes in total daily physical activity, 6 min walk test, and grip strength between arms were not significant. **Quality of life:** No significant differences in both individual arms from baseline, and no significant difference between both arms either. **Side effects:** Generally well-tolerated.
Mantovani et al., 2008 [34]	Italy	Phase III, randomized, controlled trial (*n* = 125)	Patients (aged 18 to 80 years) with malignancy of any site at an advanced stage; loss of 5% of the ideal or pre-illness body weight in the previous 3 months, abnormal values of proinflammatory cytokines, ROS and antioxidant enzymes predictive of the onset of clinical cachexia; and life expectancy of 4 months.	Medroxy-progesterone acetate (MPA) 500 mg/d or MA 320 g/d given orally as one treatment arm, or eicosapentaenoicacid (EPA) nutrition supplement 2.2 g/d, or L-carnitine at 4 g/d, or Thalidomide at 200 md/d, or MPA/MA + pharmacologic nutritional support + L-carnitine + Thalidomide	Lean body mass, phase angle, REE, daily physical activity and its associated energy expenditure, proinflammatory cytokines, fatigue, clinical response, progression-free survival, performance status, appetite, grip strength, blood levels of ROS, quality of life	**Weight gain:** Single agents were generally ineffective or mildly effective. More effective when combined (lean body mass one-way ANOVA was 1.080 ± 3.094 kg in arm 5). **Quality of life:** Significant increases in appetite (*p* = 0.003) and EQ-5DVAS score (*p* = 0.03) and an improvement in ECOG PS score (*p* = 0.03) in arm 1 (MPA or MA). Multidimensional Fatigue Symptom Inventory–Short Form, MFSI-SF (score) −2.444 ± 7.205 in arm 1. **Side effects:** Toxicity negligible and was comparable among treatment arms. Only two patients with grade 3 or 4 diarrhea were reported in arm 3 (L-carnitine) and arm 5 (MPA or MA plus EPA plus L-carnitine plus thalidomide). Overall good patient compliance.
McMillan et al., 1994 [35]	UK	Randomized, controlled trial (*n* = 26)	Patients with gastrointestinal cancer and documented weight loss of >5%, undergoing palliative therapy, life expectancy of >2 months, no surgery/RT/chemo within 2 months, no physical or functional obstruction to intake.	MA 480 mg/day, or Placebo	Weight gain, total body water, total body potassium, biochemistry and hematological parameters	**Weight gain:** No significant difference in weight gain.
McMillan et al., 1999 [36]	UK	Prospective, randomized, controlled trial (*n* = 73)	Patients with advanced or metastatic gastrointestinal cancer and documented weight loss of >5%, undergoing palliative therapy, life expectancy of >2 months, no surgery/RT/chemo within 2 months, no physical or functional obstruction to intake.	MA 480 mg/day and Ibuprofen 1200 mg/day, or MA 480 mg and placebo	Weight gain, appetite, anthropometry measurements, Karnofsky performance score, quality of life, biochemical results	**Weight gain:** After 12 weeks of treatment, there was a significant difference between the median weight gain (2.3 kg) in the megestrol acetate/ibuprofen group compared with median weight loss (−2.8 kg) in the megestrol acetate/placebo group (*p* <0.001). Accompanied by a significant decrease in the mid-upper arm circumference of the megestrol acetate/placebo group (*p* < 0.05). **Quality of life:** No significant difference in appetite changes. Significant improvement in the EuroQol-EQ-5D quality of life score of the MA/ibuprofen group (*p* < 0.05). **Side Effects:** Generally well tolerated, 3 incidents of venous thrombosis across entire study.
Navari et al., 2010 [37]	USA	Randomized, controlled trial (*n* = 80)	Patients >18 years with stage III or IV gastrointestinal or lung cancer, with weight loss >5% pre-illness, no gastrointestinal obstruction, and no major surgery, chemotherapy, or radiotherapy in the last 4 weeks.	MA 800 mg/day, or MA 800 mg/day and Olanzapine 5 mg/day	Weight gain >5%, appetite, nausea, quality of life	**Weight gain:** 15 patients on MA only gained >5% weight as compared to 33 patients on MA and Olanzapine. **Quality of life:** 2 patients on MA as compared to 25 patients on MA and Olanzapine felt that appetite improved (+3 in the visual analog scale). 5 patients on MA felt an improvement in QoL as compared to 23 patients on MA and Olanzapine. **Side effects:** Generally well tolerated.
Nelson et al., 2002 [38]	USA	Phase II Trial (*n* = 20)	Patients with anorexia due to advanced cancer excluding breast or prostate cancer and weight loss, ECOG PS 3 and below, no hormonal or chemotherapy currently.	MA 160 mg/day	Appetite, side effects, satisfaction on taking MA	**Appetite:** 15 patients reported improvement in appetite. 16 patients were satisfied with the way the medication affected their appetite. **Side effects:** Generally well tolerated, 1 case of DVT attributed to worsening disease rather than medication.
Rowland et al., 1996 [39]	USA	Double blinded, randomized, controlled trial (*n* = 243)	Patients with extensive small-cell lung cancer outside the chest or intrathoracic disease that could not be encompassed in a safe radiation treatment, ECOG 0–2, no uncontrolled infection or prior chemotherapy/RT.	Cisplatin 30 mg/m^2^/day, etoposide 130 mg/m^2^/day and, MA 800 mg/day or placebo	Benefits of MA including weight change, anorexia, n/v, Clinical response, quality of life, side effects	**Weight gain:** Weight gain of ≥10% occurred in 21% of patients in the MA group, versus only 7% in the placebo group (*p* = 0.004). **Clinical benefit:** MA associated with less nausea and vomiting compared to placebo. **Quality of life:** Minimal change in QoL and no difference over 8-month study period between the two groups. **Side effects:** Commonly includes oedema, phlebitis, thrombocytopenia and neutropenia. Grade 3/4 thromboembolic phenomenon occurred in 11 MA patients and 2 placebo patients.
Tanca et al., 2009 [40]	Italy	Phase III, randomized, controlled trial (*n* = 475)	Patients 18–80 y/o with advanced tumor at any site, loss of >5% ideal/pre-illness body weight in past 3 months, or abnormal values of proinflammatory cytokines, ROS and antioxidant enzymes predictive of the onset of clinical cachexia, life expectancy > 4 months, could be receiving concomitant chemotherapy, no significant comorbidities, no obstructive changes to body weight, no history of VTE.	A: Medroxy- progesterone 500 mg/day or MA 160 mg/BD B: Eicosapentaenoic acid with oral supplement C: L-carnitine 4 g/day D: Thalidomide 100 mg/day E: A + B + C + D	Lean body mass, resting energy expenditure, daily physical activities, IL-6 and TNF-a levels, fatigue, side effects	**Lean body mass/Weight gain:** LBM evaluated by DEXA showed a significant improvement only in arm 5 (*p* < 0.05), whilst the assessment of LBM by bioimpedentiometry did not show a significant difference in any arm of treatment. **Quality of life:** Patients in arm 5 showed a significant decrease of fatigue assessed by MFSI-SF (*p* = 0.017). **Side effects:** No toxicity of any grade was observed. Only one patient in arm 1, discontinued MPA because of DVT.
Wen et al., 2012 [41]	China	Randomized, controlled trial (*n* = 93)	Adult patients excluding women of child bearing age with advanced-stage tumor at any site, weight lost >5% of pre-illness/ideal body weight in past 3 months, life expectancy > 4 months, could be receiving chemotherapy or palliative care, no obstruction to feeding, no treatment significantly affecting weight, no previous VTE.	MA 320 mg/day and thalidomide 100 mg/day, or MA 320 mg/day only	Body weight, fatigue, quality of life, appetite, grip strength, serum levels of IL-6 or TNF-α, side effects	**Weight gain:** Both groups had significant increase in weight from baseline. Mean weight changes in trial group (with thalidomide) was significantly greater as compared to control group (*p* = 0.025). **Quality of life:** QoL and fatigue in trial group significantly improved as compared to the control group (*p* = 0.01, *p* < 0.01 respectively. A trend for improvement in appetite (*p* = 0.117) was found in the trial group as compared with the control group. **Side effects:** Toxicities included thromboembolism (3 cases), edema, somnolence and constipation at a low occurrence rate. Compliance was good.

Abbreviations: AFP, alpha fetoprotein; DVT, deep vein thrombosis; ECOG, Eastern Cooperative Oncology Group; HCC, hepatocellular carcinoma; LBM, lean body mass; MA, megestrol acetate; MFSI-SF, Multidimensional Fatigue Symptom Inventory-Short Form; RCT, randomized, controlled trial; RT, radiotherapy; QoL, quality of life; SD, standard deviation; VTE, venous thromboembolism.

## Data Availability

The datasets generated during and/or analyzed during the current study are available from the corresponding author on reasonable request.

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
