# Peer review of "A Systematic Review and Meta-Analysis of the Clinical Use of Megestrol Acetate for Cancer-Related Anorexia/Cachexia"

_jcm, 2022, doi:10.3390/jcm11133756_

Round 1

Reviewer 1 Report

The authors have conducted a meta-analysis to evaluate the potential clinical benefits of megestrol acetate in patients with cancer related anorexia.  This provides a significant update compared to prior reviews.  The main endpoint that could be evaluated was mean change in weight from this analysis.  Even with high dose treatment, weight loss was more common than weight gain and studies were insufficient to evaluate skeletal muscle endpoints or quality of life.  The authors concluded that megestrol acetate was not effective in providing improvement in anorexia in patients with advanced cancer.  While this information is not new, the authors have done an excellent job of tabulating each of the studies and their key findings, which is quite valuable to those in the field of anorexia and cachexia.   This information should also be helpful for those involved with guideline development for patients with anorexia or cachexia.  No suggestions for revision.

Author Response

Thank you for the very positive comments and encouragement!

Reviewer 2 Report

Authors report a complete and exaustive review concerning the use of megestrol acetate in cancer related anorexia. Prior to acceptation authors should use the term anorexia/cachexia instead of anorexia. Moreover, some point should be added to the paper concerning the importance that nutrition in cancer patients has gained to day as testified by the nutritional algorythm published in the last years 

Author Response

Thank you for the comments.

  1. We have now used the term "anorexia/cachexia" instead of "anorexia".
  2. We have added a comment in the introduction as per your suggestion, "Providing dietary counseling, nutritional support and nutritional therapies are therefore important and endorsed by major clinical practice guidelines (Arends et al., 2021)."

This manuscript is a resubmission of an earlier submission. The following is a list of the peer review reports and author responses from that submission.